# Drought-Induced Challenges and Different Responses by Smallholder and Semicommercial Livestock Farmers in Semiarid Limpopo, South Africa—An Indicator-Based Assessment

**Leonhard Klinck** [1,2,*], **Kingsley K. Ayisi** [2] and **Johannes Isselstein** [1]

1   Department of Crop Sciences, Grassland Science, Georg-August-University, 37073 Gottingen, Germany; jissels@gwdg.de
2   Risk and Vulnerability Science Centre, University of Limpopo, Sovenga 0727, South Africa; kwabena.ayisi@ul.ac.za
*   Correspondence: leonhard.klinck@uni-goettingen.de

**Abstract:** Increased seasonal climatic variability is a major contributor to uncertainty in livestock-based livelihoods across Southern Africa. Erratic rainfall patterns and prolonged droughts have resulted in the region being identified as a climate 'vulnerability hotspot'. Based on fieldwork conducted in the dry seasons in a semiarid region of South Africa, we present an interdisciplinary approach to assess the differential effects of drought on two types of livestock systems. Organic matter digestibility, faecal crude protein, C/N ratio and the natural abundance of faecal $^{15}$N and $^{13}$C isotopes were used as ecophysiological feed quality indicators between smallholder and semicommercial systems. These measurements were complemented with qualitative surveys. In a novel approach, we tested the potential of the isotopic signature to predict feed quality and present a significant relationship between organic matter digestibility and isotopic ratios. Indicators assessed smallholder feed quality to be significantly higher than semicommercial feed. However, animals from semicommercial farms were in significantly better condition than those from smallholding farms. Differential access to feed resources suggests that a complex feed–water–land nexus pushes smallholders into high reliance on off-farm supplements to bridge drought-induced feed deficits. The paper thus offers a contribution to intersectional work on drought effects on livestock keepers of semiarid South Africa and illustrates how ecophysiological indicators mirror socioeconomic differences.

**Keywords:** communal rangelands; digestibility; drought; faecal N; forage quality; supplementary feeding

## 1. Introduction

Across Southern Africa, rainfed mixed crop–livestock systems are the main agricultural practices among both smallholding and more commercialized farmers [1]. Environmental stress, caused by events such as droughts, affects rangeland productivity directly, and this has consequences in terms of reduced physiological wellbeing and reproductive health among animals, which in turn threaten the livelihoods of farmers in affected regions [2–5]. In Southern Africa, cattle, in particular, are not regarded as just an economic asset; they fulfil multi-faceted roles, especially in communal areas, ranging from direct, agriculture-related provisions (draught, manure, milk, meat) to a myriad of sociocultural functions [6–10]. Thus, seasonal, drought-induced feed deficits lead to various downstream consequences both for animals and farmers. The biophysical drought–feed deficit nexus is well documented for other semiarid contexts, e.g., [11–15], whereas in South Africa, scholarly work has placed greater emphasis on ex post assessments of political drought interventions [16–18]. One such intervention is the drought relief scheme administered by the National Disaster Management Centre [19]. These are reactive approaches by the state to mitigate the drought-induced livestock loss of resource-constrained farmers and entail the provision of supplementary feed or vouchers as financial support.

In this study, we aim to contribute to an indicator-based assessment of drought and its effects on grazing livestock, which form the basis of many people's livelihoods across Southern Africa. Specifically, we take drought as a starting point for inquiry into the severity of feed shortages, linking biophysical indicators and farmers' perceptions. While indicators from ruminant ecophysiology are useful proxies to quantify the severity of the effects of environmental stresses such as drought, a qualitative approach accounts for differences in context and enables information on farmers' subjective perceptions to be captured. First, body condition scoring (BCS) enables an estimation of the individual animal's health and dietary situation by assessing muscular and fatty deposits in a rapid, non-invasive way [20]. Second, the digestibility of organic matter (OMD) and the faecal carbon to nitrogen (C/N) ratio derived from faecal N and organic matter (OM) are common indicators for the nutritional and energy value of ingested feed [21–27]. The prediction of feed organic matter digestibility (OMD) is based on a direct relationship between feed-N and faecal N [22,23,28,29]. Rangeland forage is more acceptable and likely to have greater intake by grazing ruminants when consumed during early maturation stages, when there is a relatively narrow C/N ratio. Feed quality decreases during maturation and excessively long dry periods, which results in increased concentrations of lignin and other indigestible fibre constituents, as well as a widening of the C/N ratio [30]. Thus, low forage N is typically associated with poor-quality forage, leading to low nutrient intake in grazing animals, as well as low overall forage intake and long digestive retention time [31]. Dietary C/N content is related to faecal content in cattle, as protein-rich feed provides more growth substances for intestinal microbes, thereby increasing faecal bacterial biomass [30,32,33]. Faeces from cattle fed poor-quality feed thus typically show a higher C/N ratio than faeces from high-quality feed.

We also propose a third indicator for drought stress in the rangeland-based contexts of Southern Africa by linking OMD to the natural abundance of $^{15}N/^{14}N$ and $^{13}C/^{12}C$ isotope ratios in cattle faeces and present a relationship where OMD responds significantly to the isotopic signature. In the biosphere, geosphere and atmosphere, C and N occur as isotopes with different masses in different absolute abundances. As one isotope may become enriched or depleted over the other (fractionation), the ratios between light and heavy isotopes vary in their environment. For example, different accumulation rates of the two C isotopes in plants occur with $C_3$-plants containing less $^{13}C$ than $C_4$-plants [34]. In ruminants, the isotopic composition of ingested biomass is then subject to further fractionation due to metabolic, physiological and microbial processes. As Southern African savannah monocots are dominated by $C_4$-plants, and shrubs and forbs are commonly $C_3$-plants [35], these distinct carbon isotope compositions are reflected in the faeces of ruminants that consume these different plant forages, thereby providing a useful indicator for short-term dietary reconstruction [36–40]. Feed nitrogen fractionation in ruminants occurs in the rumen and during intestinal digestion. Due to the mass differences, $^{15}N$ and $^{14}N$ isotopes have different reaction rates in biotic processes, where $^{14}N$ is depleted faster within the N-pool. This is also the case for the digestion pathway from ingestion to excretion in mammals, where the accumulation of $^{15}N$ and the depletion of $^{14}N$ occurs due to intestinal microorganisms that constitute a higher trophic level, e.g., [38,40,41]. We expect OMD to correspond to faecal C and N isotope ratios in that high faecal $^{13}C$ content predicts low OMD and, conversely, high faecal $^{15}N$ predicts high OMD. Thus, the information previously derived from isotope analyses could be extended to draw conclusions on feed digestibility as a second approach to feed OMD in semiarid rangeland-based diets. We assume that comparisons of faecal isotopic signatures are a potential indicator to assess the nutritive feed value, and we present the first results of a positive correlation between OMD and faecal $^{15}N$ and the inverse relationship for OMD and faecal $^{13}C$. This case study as part of the ongoing research project SALLnet ([42]) and draws from fieldwork on private ranches and communal rangelands in semiarid South Africa. The work complements information on the above indicators with social perceptions to elicit the magnitude of drought–feed

deficits and the differential perceptions of challenges related to drought between two social classes of livestock farmers.

## 2. Materials and Methods

Recognizing the potentially artificial dichotomy of framing socioeconomic farm classes as 'smallholder' (SH) and 'semicommercial' (SC) [43,44], our stratification follows the official classification based on categories such as land ownership and, therefore, access to private rangelands, herd size, material assets and the motivation of livestock keeping [9,45,46]. Access to resources affects the ability of farmers to deal with hazardous environmental events like droughts [47] and is therefore central to our stratification, where SC drew from private rangelands and SH farmers relied on communal rangelands as their main grazing resource. The farm surveys followed a nonprobability method [48], where the farmers within our two population subgroups were purposively selected with the help of local state extension officers based on the classification criteria. Farms were either mixed crop–livestock systems or systems with livestock only. Fieldwork included ecological biomass sampling, individual open-ended questionnaires and group discussions. In total, 30 farms were surveyed (Table 1) from September to November in 2018 and 2019. This period typically corresponds to the period of transition from the dry to rainy season in Southern Africa.

**Table 1.** Characteristics of selected semicommercial and smallholder farms in two semiarid municipalities of Limpopo.

| Farm Class | Grazing Resource | Number of Farms | Stocking Density |
|---|---|---|---|
| Semicommercial | Fenced ranch | Total: 11 | Min.: 0.06 LSU/ha |
| | <500 ha | 5 | Max.: 1.85 LSU/ha |
| | <1000 ha | 3 | |
| | ≥1000 ha | 3 | |
| Smallholder | Communal rangelands | Total: 19 | |

### 2.1. Study Sites

The study was conducted in Limpopo, South Africa's northernmost province, which has subtropical and more humid conditions only in mountainous areas. Farmers across the municipalities of Maruleng and Lephalale were selected as these sites are in some of the driest parts of the province. The selection was based on the classifications of agroecological zones [49–51], where the mean monthly temperature exceeds 18 °C throughout the entire year and the length of the growing period (LGP) is less than 70 days. LGP is defined by the period of the year where temperatures are equal to or greater than 5 °C and the sum of soil moisture and precipitation exceeds half the potential evapotranspiration, thereby favouring crop growth under natural conditions [51]. Based on official livestock census data, most cattle farmers are smallholders on communal land (95%) owning, on average, 9 head of cattle. In marked contrast to these communal, livestock-keeping farmers, homesteads or villagers, the farmers of the landholding, (semi)commercial sector, have, on average, 79 head of cattle per farmer [52], amounting to roughly a third (29.7%) of the total number of cattle (Figure 1). Nguni–Brahman crossbreeds are the most popular cattle type in both SH and SC farming systems.

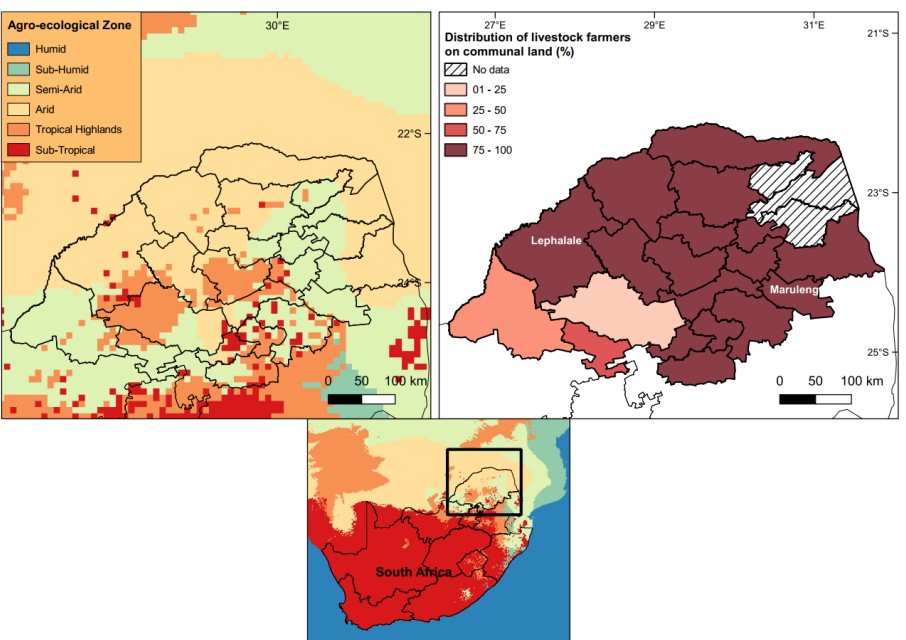

**Figure 1.** Left: Agroecological classification of Limpopo, based on data from HarvestChoice [50]. Right: Proportion of communal compared to commercial cattle farms across Limpopo based on census data [52]. Sampling took place in the municipalities of Lephalale and Maruleng. Inset (below) shows the location of the study area and the agroecological classification of Southern Africa. Own map, created using the Free and Open Source QGIS.

### 2.2. Social Subsystem

Open-ended interviews captured the perceived challenges in meeting dry season livestock feed demand and were conducted in an exploratory manner free from preconceived notions of drought and its potential effects on people's livelihoods. Responses were recorded and recurring themes were identified during content analysis. In addition, three group discussions with both the previously surveyed farmers and local extension officers were conducted with 22, 10 and 13 participants, respectively. The main goal of these discussions was to learn about both responsive strategies with which farmers respond to the challenge of recurring feed deficits. All farm visits and group discussions included informed consent from all attendees, simultaneous translation by a native Northern Sotho-speaking research assistant and were facilitated by a local agricultural state extension officer who mediated between researchers and farmers.

### 2.3. Ecophysiological Subsystem

Each animal was photographed, and the physical condition was assessed through body condition scoring, which ranks animals from 0 (emaciated) to 5 (overfat) [20,53]. Faecal OM was determined by subtracting ash, obtained through combustion from the total dry matter of faecal samples (OM = DM − ash). Crude protein (CP) was derived using the multiplication conversion factor of 6.25 developed by FAO/WHO [54] and which has been shown to provide an adequate approximation of (tropical) grasslands [55]. Feed OMD was determined based on the formula developed by Lukas et al. [22]:

$$\text{OMD} = 79.76 - 107.7e^{-0.01515x} \text{ where x = g CP/kg faecal OM} \tag{1}$$

From the 30 sampled farms, four fresh faecal samples per farm from defecating animals were collected either from heifers, adolescent bulls or nonlactating cows to ensure comparability. The samples were dried in a forced-air oven at 60 °C until constant weight was achieved, ground through a 0.2 mm sieve and analysed for N and C fractions at the Centre for Stable Isotope Research and Analysis in Göttingen, Germany, via isotope mass spectrometry (mass spectrometer: Delta Plus, Finnigan MAT, Bremen, Germany;

elemental analyser: NA1110, CE—Instruments, Milano, Italy). The ratio of heavy and light isotopes per sample was measured and determined against the ratio contained in a primary reference substance. For C, this was V-PDB ($^{13}C/^{12}C = 0.0111802$), and N was measured against the composition in atmospheric $N_2$ ($^{15}N/^{14}N = 0.0036765$). The laboratory standard calibrated against the primary standard was acetanilide ($C_8H_9NO$). The relative difference between sample and laboratory standard ratios is expressed as a delta (δ)-value and used as the unit to express the relative abundance of isotopes in a sample relative to the abundance in the standard:

$$\delta(‰) = \left( \frac{Rsample}{Rstandard} \right) - 1 \times 1000 \tag{2}$$

Descriptive statistical procedures were performed in R version 4.0.5 [56]. Differences between means in non-parametric data were compared with Welch's ANOVA when variances were heteroskedastic and with the Mann-Whitney U-test in the case of homoskedasticity. Differences between normally distributed but heteroskedastic data were also compared with Welch's ANOVA. Means of normal and homoskedastic data were compared with a two-sample *t*-test. For all tests and linear models, a significance level of $\alpha = 0.05$ was assumed.

## 3. Results

Both semicommercial and smallholder farmers perceived feed deficits to generally be the most severe during the late dry season, with October being most critical, while late summer months are characterized by feed abundance (Figure 2). Farmers in the SH class reported the period of observed weight loss in cattle to be one month longer than SC farmers (3.4 vs. 2.4; data not shown).

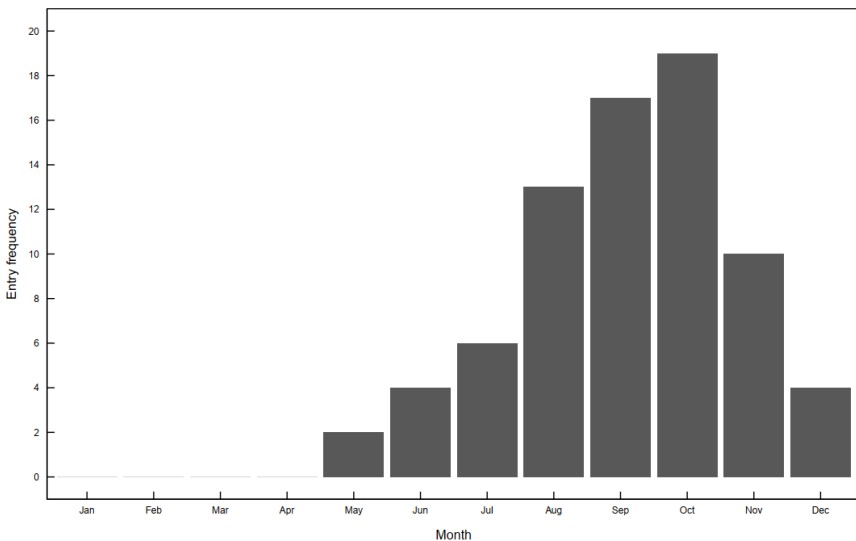

**Figure 2.** Intra-annual periods of cattle feed shortages as perceived by all (n = 30) farmers. Feed shortages peak in early spring (September–October) but are entirely absent during the summer and autumn months (January–April).

### 3.1. Human Subsystem: Differential Challenges and Responses during Drought Feeding

Within the social subsystem, the perceived challenges related to livestock feed supply differed between the two farm classes, but farmers in both classes expressed 'drought' to be a common challenge. Responses to open-ended questions and results from group discussions were grouped into themes. This resulted in four main root causes in each of the two farm classes. Respondents elaborated on further associated consequences directly affecting the feed supply situation and on their adaptive responses to drought-induced feed shortages. This causal chain was conceptualized in a concept map depicting root causes, intermediate effects and adaptive responses as deemed by respondents (Figure 3).

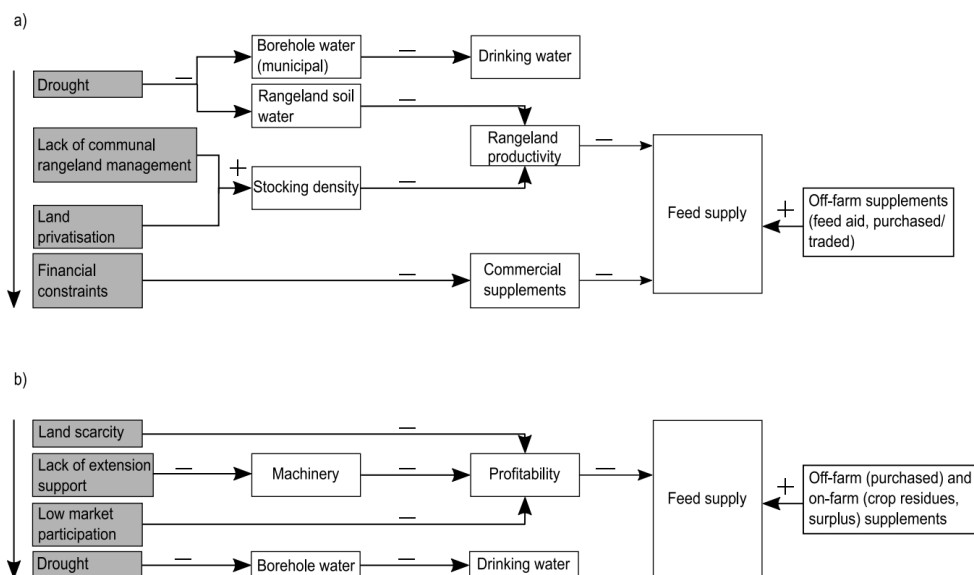

**Figure 3.** Concept map of perceived differential root causes (grey) and consequential, intermediate causes (white) for dry season cattle feed shortages among (**a**) smallholder and (**b**) semicommercial livestock farmers. Vertical arrows on the left indicate root causes listed in descending order of weight, as deemed by the respondents. Horizontal, relational arrows indicate enhancing (+) or reducing (−) effects on conceptualized intermediate causes.

Farmers in the SH class referred to water shortages as the most frequently perceived challenge to livestock (Figure 3a). Respondents revealed a causal chain where 'drought' was reported to translate into two major downstream consequences: a lack of forage availability and inappropriate municipal and borehole water supply, a crucial requirement when animals are kept in a village homestead. A communal farmer from Lephalale stated

> *'The taps are almost always dry. For us to get the water in the morning, it can last maybe, if you're lucky, three hours and [ . . . ] many people don't have boreholes, they're just relying on this municipal water to make sure they feed water to the animals.'*

As a result of dry communal dams, farmers were either forced to buy water from the municipality or walk their animals to alternative water sources further away. Among SC farmers, meteorological drought was the least frequently perceived challenge or root cause of feed deficit, although respondents also experienced water shortages primarily affecting the availability of borehole water supply capacity and, thus, drinking water. Another central theme among SH farmers' perceived challenges to feed supply was poor rangeland management associated with the absence or destruction of grazing camps and a perceived lack of boundaries for controlled grazing. Land was perceived to be insufficient to support the number of cattle. Access to land was perceived to be insufficient to support the number of cattle and was related to the expansion of cultivated land at the expense of grazing areas. In the words of an SH livestock keeper:

> *'There is a lot of people that are just ploughing the fields. The grazing fields are dwindling. Every year the grazing space is declining.'*

Together, these causes appeared to limit access to sufficient grazing land and were thus conceptualized as main drivers of high grazing pressure due to high stocking density and declining rangeland productivity. For SC farmers, land was perceived to be the greatest of all limitations, although it was seen as a resource constraint limiting farm expansion and profitability (Figure 3b) rather than compromising their animals' nutritional requirements. Other challenges associated with feed shortages in SC farming systems included a perceived absence of governmental technical extension services supported with machinery for mowing and forage preservation. High transaction costs in livestock

production included large distances to commercial hubs and cattle auctions and little bargaining power with retailers. In this sense, limited market participation and profitability were other challenges unique to SC farmers affecting the cattle feed supply situation. Notably, all farmers reported the need to bridge feed shortages with both purchased supplements and surplus produce from various cash crops.

### 3.2. Ecological Subsystem: Differential Severity of Feed Deficits

In the ecological subsystem, 119 faecal samples from heifers, adolescent bulls or non-lactating cows were collected from n = 74 SH and n = 44 SC farms. Feed quality indicators expressed as OMD, faecal C/N ratio and faecal CP differed significantly between the two farm classes (Figure 4a–c): the OMD of SH feed was significantly higher on average than the OMD of SC farmer feed, $t(116) = 4.69$, $p < 0.001$. The Mann–Whitney U-test revealed C/N ratios to be significantly higher in faeces from SC cattle ($p < 0.001$, effect size $r = 0.41$). Mean faecal crude protein was higher in SH cattle; Welch's $F(1,115.46) = 23.96$, $p < 0.001$. Figure 4d shows the difference in the mean body condition of animals, with SH cattle being in a significantly poorer condition than SC cattle; Welch's $F(1,69.85) = 38.98$, $p < 0.001$.

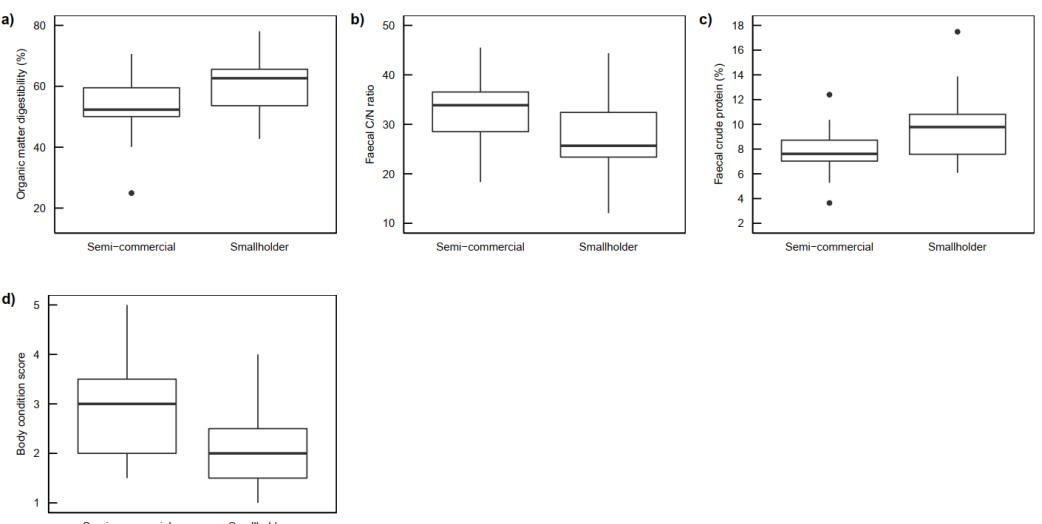

**Figure 4.** Comparison of (**a**) feed organic matter digestibility, (**b**) faecal C/N ratio, (**c**) faecal crude protein and (**d**) body condition scores between individual cattle from smallholder (n = 74) and semicommercial (n = 44) farms.

In both farm classes, a simple linear regression predicted OMD based on the $\delta^{13}C/^{12}C$ ratio (Figure 5a). In SH cattle, the relationship was more significant ($F(1,72) = 110.1$, $p < 0.001$, $R^2_{adj.} = 0.5992$) than in SC cattle ($F(1,42) = 6.444$, $p < 0.05$, $R^2_{adj.} = 0.133$). Predicted OMD percentages were equal to $24.5625 - 1.6919\,(\delta^{13}C/^{12}C \text{ ratio})$ % and $30.026 - 1.302\,(\delta^{13}C/^{12}C \text{ ratio})$ %, respectively. In contrast, OMD correlated positively with increasing $\delta^{15}N/^{14}N$ ratios (Figure 5b). The relationship was again more pronounced in SH cattle ($F(1,72) = 11.82$, $p < 0.001$, $R^2_{adj.} = 0.1291$) than in SC cattle ($F(1,42) = 5.689$, $p < 0.05$, $R^2_{adj.} = 0.09833$), and the predicted OMD percentages were equal to $49.8495 + 1.9119\,(\delta^{15}N/^{14}N \text{ ratio})$ % and $45.1997 + 1.7927\,(\delta^{15}N/^{14}N \text{ ratio})$ %, respectively.

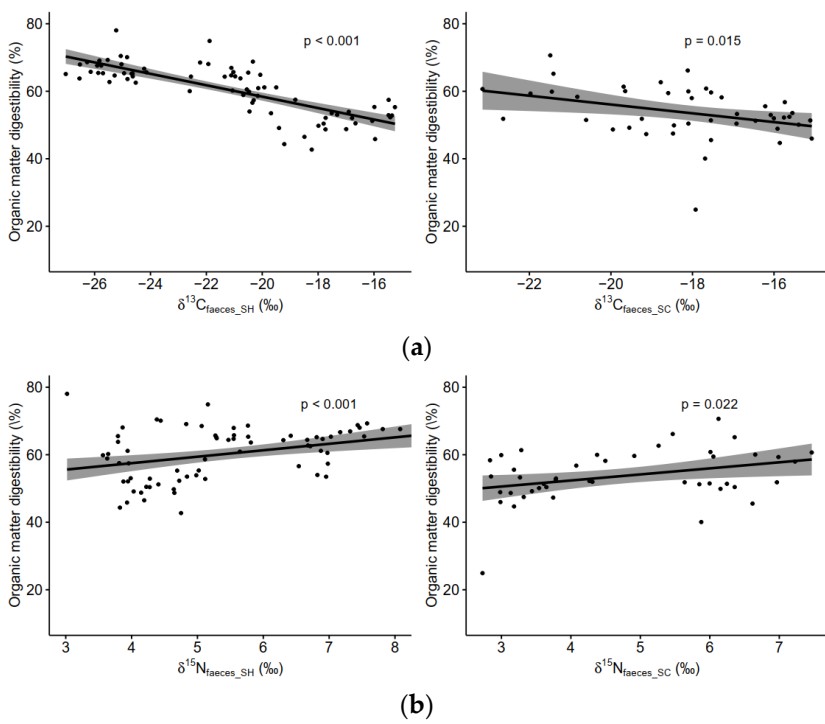

**Figure 5.** Relationships between the faecal isotopic signature and feed organic matter digestibility for (**a**) $^{13}C$ and (**b**) $^{15}N$ content, with similar results in both farm classes. Linear regressions were significant to highly significant at moderate to low goodness-of-fit ($R^2$).

## 4. Discussion

The study does not claim to account for a comprehensive farm system understanding, including the causes and mechanisms for the differential vulnerability of rural livestock systems to drought. Rather, it offers a 'snapshot of vulnerability at one point in time' [57] and not a representation of a year-round pattern in the dietary regime. Interviews and discussions laid out the temporal extent of feed deficits across all farmers (Figure 2) as well as the differential challenges within the drought–feed nexus. SH farmers and communal land users, in general, are less resilient towards feed shortages [58], a consequence attributed to the poor governance of resource-constrained rural South African municipalities [43,59–61]. In periods of drought, basic requirements like water and feed were experienced as the greatest limitations. As shown by Lamega et al. [37], smallholder farmers primarily resort to feed purchase (or using on-farm crop residues, where applicable) and herd size reduction as coping strategies during periods of drought-induced feed shortages. Feed aid schemes played a minor role in the farmers' perceptions. The information summarized in Figures 3a and 4d, however, suggests that these strategies are incapable of averting livestock health decline and starvation. In light of the smallholders' views and the physiological condition of most of their animals (Figure 4d), governmental feed aid under the existing arrangements does not alleviate these limitations effectively. Results further concur with the studies of Martin et al. [62] and Vetter et al. [63], who demonstrated how droughts translate into different downstream consequences directly affecting rangeland productivity.

Generally, in cattle, protein-rich feed has a positive effect on degradability and rumen fermentation. High OMD and low C/N ratios are thus usually associated with high-quality feed [33]. The significantly higher feed digestibility (Figure 4a,b) in SH farming systems may therefore seem to contradict the poor physiological state of SM animals, especially in light of the better socioeconomic and physiological conditions of SC animals. Considering physiological dynamics, our findings suggest that high-protein supplements in SH farming systems play a pivotal role in their dry season feed regimes, where very low feed intake levels in SH animals are characterised by a high proportion of protein-rich supplementary feed. This is likely because access to natural forage is severely limited by insufficient access

to high-quality pastures and the poor nutritional quality of available forage. The high proportion of supplementary feeding, however, appears to be barely sufficient to meet the maintenance energy requirements of livestock. Given that very low feed intake levels further reduce OMD in the rumen physiologically [64–66]—even when extra protein is added [67]—it can be assumed that both the forage quantity and quality of communal rangelands from the selected sites are extremely poor. This assumption was validated by Lamega et al. [37] in forage and feed value analyses from other communal rangelands in Limpopo. On the other hand, slightly reduced feed intake at 80% of the maintenance energy requirement increased OMD, and only further reductions led to decreased OMD [68,69]. This is likely due to lower rumen fermentation and digestion efficiency for fibres at such low intake levels, rather than being due to the feed nutritional properties themselves. Therefore, we caution against using OMD and C/N ratios as feed quality indicators for an accurate representation of the momentary dietary situation of cattle, as these indices may not account for quantitative intake levels.

On the other hand, the high faecal C/N ratio in SC farmers' cattle (Figure 4b) suggests a lower proportion of protein-rich supplements in their diet. The snapshot of the dry season feed regime points to a rangeland-based diet with a high proportion of $C_4$ grasses—including residues from maize (*Zea mays*)—with typically high C/N ratios and, thus, lower digestibility [30,70,71]. The high BCS is thus likely due to dietary diversification with the supplementation of crops and crop residues. This may also explain both the less pronounced relationship between the isotopic signatures and OMD and the relatively high variability in SC cattle. Because of the indication of lower OMD compared to SH cattle, we assume off-farm supplements, such as concentrates, play a minor role in the overall dry season feed regime. Access to fenced, managed rangelands and the possibility to preserve fresh forage appear to constitute a crucial resource for meeting the energetic requirements of cattle in a drought-affected, semiarid Southern African context. High digestion efficiency in the rumen may be another cause of depressed OMD levels. Feeding high-quality forage at or above maintenance level was previously shown to increase the rumen passage rate of particles, thereby reducing rumen retention digestion efficiency [64,72]. Chaokaur et al. [73] and Gabel et al. [74] found increased feeding levels to even depress digestibility. We assume this effect to be mirrored in our results. Cerling and Harris [75] and Cerling et al. [76] demonstrated how the study of C isotope ratios in animal tissue is particularly useful in Sub-Saharan Africa to derive information on a ruminant's dietary composition. The significant correspondence of OMD to $\delta^{13}C/^{12}C$ ratios supports the assumption that rangelands offer predominantly $C_4$ monocots of poor quality to animals during the dry season. Given the model's moderate capacity to explain variability, further studies need to specifically investigate the effect of plant maturity stages and supplementary high-quality $C_4$-feeds—for example, maize silage—in rangeland-based dry-season feed regimes, as they likely affect the model's predictive capacity. The accumulation of $^{15}N$ in faeces is due to intestinal, microbial growth that depletes the lighter $^{14}N$ in the digesta [38,40,41] and, although losses occur, the feed–faeces fractionation rate is assumed to be consistent [77]. Therefore, high-protein feed generally results in elevated levels of faecal $^{15}N$ [78]. The observed high feed variability in a rangeland-based diet may include the increased ingestion of antinutritional plant parts [79], which, in turn, results in reduced correlations between faecal protein [25]—and, thus, $^{15}N$—and OMD.

Cognisant of the drought stress situation, we consider locally tailored, practical feeding strategies to be promising alternatives to scheme-based governmental provisions. Agronomic strategies may consider the nutritive potential of winter (drought) forage grasses [80], legume residue silage [81], legume trees [82] or other woody browse species [83,84] as supplementary feed in addition to poor-quality forage. Fundamental, however, is the transfer of such approaches from sustainability science to people's contextual realities and we thus concur with other authors, e.g., [85–87], who highlight the need for transformative pathways recognizing the inclusion of stakeholder interests and participation. Controversies in feed aid schemes were also discussed by Müller et al. [88], who concluded

that feed aid was effective as a short-term response but compromised options to adapt to drought in the long term. The SC farmers appeared to respond much more flexibly to drought-induced feed gaps and are thus better positioned to adapt. A strong reliance on off-farm supplements would be problematic for smallholding livestock keepers in two ways: First, high expenditures on feed supplements thwart the function of livestock as 'invisible capital' [89] and literally eat up the basic rationale of keeping cattle as a store of wealth; secondly, the financial pressure of feed expenses may increase the risk of creating a cycle preventing already constrained SH households from investing and engaging in economic activities beyond agriculture [90]. This risk is to be considered if multisectoral rural development policies are to be developed [9,10,85] and farming system flexibility [91] achieved as a prerequisite for risk adaptation. We thus highlight Vetter et al. [63] and Atkinson's [43] calls for integrated, nuanced approaches that involve a strengthening of local institutions and land stewardship but also maintain flexibility for the multitude of Limpopo's types of livestock-based livelihoods. Finally, as water lies (or flows) at the heart of animal welfare within the drought–feed nexus, its poor public distribution by municipalities quickly aggravates the situation, especially for SH farmers. A complex configuration of controlling municipalities, high climatic variability, different farming systems, land rights, watersheds, irrigation and drinking water infrastructures are all elements of local 'hydrosocial territories' [92] that cannot be treated as isolated from livestock performance and feed supply. Drought and its multifaceted consequences for SH farmers thus reduce flexibility for adaptive responses and are likely to extend the evident physiological feed gap to a wider social gap between livestock keepers.

## 5. Conclusions

Different social classes of livestock keepers, who are equally exposed to meteorological drought, differed in their responses to feed shortages. Semicommercial farmers were shown to operate under socioeconomic conditions that allow for more flexibility and adaptive responses within the feed regime. This observation was made in a multidisciplinary, indicator-based approach that drew from surveys conducted in the social subsystem and feed value indicators in the ecophysiological subsystem. Feed organic matter digestibility, faecal C/N ratios and body condition scoring proved to be useful proxies to mirror the recent qualitative intake in ruminants, but they are less useful as indicators of the quantitative intake of matter under situations of high feed variability and uncertainty. For rangeland-based diets in semiarid environments, we revealed a significant relationship between faecal $^{15}$N and $^{13}$C and feed value. The findings feed into a broad body of research on measurement techniques in ruminant nutrition [93] and may extend it by a new indicator. We thus propose to further investigate the capacity of isotopic signatures to predict feed organic matter digestibility. Recognizing the study's limitations in its spatial and longitudinal scope, we believe that the results have revealed relevant insights into the drought–feed deficit nexus in Southern Africa.

**Author Contributions:** Conceptualization, L.K. and J.I.; formal analysis, L.K.; funding acquisition, J.I.; investigation, L.K.; methodology, J.I.; project administration, K.K.A. and J.I.; resources, K.K.A. and J.I.; supervision, K.K.A. and J.I.; validation, K.K.A. and J.I.; visualization, L.K.; writing—original draft, L.K.; writing—review and editing, L.K. and J.I. All authors have read and agreed to the published version of the manuscript.

**Funding:** The study was conducted within the SPACES2-SALLnet project (grant number 01LL1802A), funded by the German Federal Ministry of Education and Research (BMBF).

**Institutional Review Board Statement:** The study was conducted in accordance with the Declaration of Helsinki and approved by the Ethics Committee at the University of Limpopo.

**Informed Consent Statement:** Informed consent was obtained from all subjects involved in the study. Before the survey and on-farm sampling work research aims and procedures were explained to interviewed persons by extension officers and committee members entitled by the University of Limpopo. This offered participants the opportunity to ask questions and discuss the research in their

own language. Participants were reminded of the conditions before each interaction and asked again for consent.

**Data Availability Statement:** The data that support the findings of this study are available upon request from the authors.

**Acknowledgments:** We thank the farmers for extensively sharing their time and their willingness to participate in the surveys, as well as the local extension officers for their coordinating role as survey supervisors and translators. Without their essential role in facilitating access to farms and households, this study would not have been possible. We thank the supporting teams of the involved South African and German departments for technical and logistical assistance, especially Kabisheng Mabitsela from the Risk and Vulnerability Science Centre, University of Limpopo, South Africa, for essential assistance in the field and translation. We further thank Alan Hopkins from the Centre for Rural Policy Research at the University of Exeter for English language editing.

**Conflicts of Interest:** The authors declare no conflict of interest. The funders had no role in the design of the study; in the collection, analyses, or interpretation of data; in the writing of the manuscript; or in the decision to publish the results.

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
