# Peer review of "Drought-Induced Challenges and Different Responses by Smallholder and Semicommercial Livestock Farmers in Semiarid Limpopo, South Africa—An Indicator-Based Assessment"

_sustainability, doi:10.3390/su14148796_

Round 1

Reviewer 1 Report

The Ms “Drought-induced challenges and different responses by smallholder and semi-commercial livestock farmers in semi-arid Limpopo, South Africa – an indicator-based assessment” is very nicely designed and executed. It is a unique study that should be published.

The Ms is also very nicely written. Authors have discussed almost each aspect of their findings. Abstract and conclusion is also very well written.

Only a few minor changes are required-

1.      Author should mention the source of figure 1.

2.      Table 1, SC and SH should be elaborated.

3.      Figure 2, error bars with statistical data should be included.

4.      Figure 3 and 4 are not very clear. It should be replaced with better figures.

5.      Figure legends should be elaborated in each figure. 

Reviewer 2 Report

Line 111: better describe the methodology proposed by Shively (2011), since it will be an important analysis for the papper.

Line 131: Figure 1 - Distribution of livestock farmers, I suggest changing the gray scale to a color scale and also inserting the legend for NA (non apply? Protection area?)

Line 136: Table 1 - What is LSU (stocking density)? Insert in table caption

Line 185: Why the weight loss in cattle to be one month longer than SC farmers? Did the SC farmers offer any food supplements in the cattle's diet?

Line 209: The unreliable water provided by the municipality is a socio-political problem that aggravates the situation and must be treated as critical and fundamental in compromising the cattle diet and consequently the family income of farmers. The scarcity of water ravages the world and the treatment of drinking water should be a priority in any country, regardless of the continent. How could the authors use the work to highlight such a lack of responsibility on the part of the municipality's water supply system? Was any bromatological test performed on this supplied water? The word "unreliable" opens room for defense of the municipality, that is, being dubious does not mean being inappropriate for animal consumption, right? I suggest removing this word in case of absence of any analysis of the quality of this water or replacing it with the word "inappropriate", directly accusing the municipality's negligence with the directing of the collected taxes/taxes for the treatment of the water of its taxpayers.

Line 214: Is this water the same as mentioned above (line 209)? If the farmers pay for it, they have the right to demand that it be drinkable, don't they? I'm confused now, in my country all water supplied is paid for and therefore it must be drinkable. This water is drinkable, right?

Line 226: Why the grazing fields are dwindling? Any agricultural or urban expansion? You need to explain why, right?

Line 233: I agree with the lack of technical extension support from the government, but how could the provision of agricultural machinery contribute to the forage preservation? I see here, and it was even indicated in the text above, that forage are poorly maintained due to incorrect livestock practices, such as the excessive number of heads of cattle per hectare, in association with other factors such as type of forage, availability of inputs/nutrients , space management, among others. Such practices are enhancing the quantity/quality of pasture. The government should offer technical assistance in this regard (improve the forage management). Proper management and recovery of degraded pastures are emergency measures for ranchers to avoid the migration of cattle to areas with water availability and ensure greater bargaining power with retailers.

Line 281: I know this is beside the point but what breed of cattle are normally raised in Limpopo? An immediate alternative would be to replace it with a more rustic breed (eg.: Sindi cattle, from Pakistan) in the harsh conditions of the region. If the environmental and economic conditions do not favor the raising of cattle, be it of any breed, who knows the farmers are insisting on raising cattle where caprines should be raised?

Line 267: Discussing the excellent and rich work, what I want here, taking the scientific discussion to another scope, is to open an mind to return to the farmers as an option to improve the quality of life of the rural family and its cattle. When government support does not reach small farmers and the transforming source of everything (water) is not enough to change the nutritional quality of the herd, it is better to think about modifying the activity instead of witnessing the cattle languishing in the pasture and not finding a solution to the problem. Changing thinking is sometimes more difficult than making it rain in the semi-arid region or getting help from governments that are negligent towards their people.

Reviewer 3 Report

Increased seasonal climatic variability is a major contributor to uncertainty in livestock based livelihoods across Southern Africa. Erratic rainfall patterns and prolonged droughts have resulted in the region being identified as a climate vulnerability hotspot’. Based on fieldwork conducted in the dry seasons in a semi-arid region of South Africa, we present an interdisciplinary approach to assess the differential effects of drought on two types of livestock systems. Organic matter digestibility, faecal crude protein, C/N ratio and the natural abundance of faecal 15N and  C 13 isotopes were used as ecophysiological feed quality indicators between smallholder and semi-commercial systems.  We think that this work has high potential for publishing in the journal. We ask only the authors to add some more recent related work in this area (for example,doi.org/10.3390/su13094607 and doi.org/10.3846/jeelm.2022.15483) and re-control the details of the figures in page 8. 
